# Influence of energetic particle precipitation on Antarctic stratospheric chlorine and ozone over the 20th century

Ville Maliniemi[1], Pavle Arsenovic[2,1], Annika Seppälä[3], and Hilde Nesse Tyssøy[1]

[1]Birkeland Centre for Space Science, Department of Physics and Technology, University of Bergen, Norway
[2]Risk Management Solutions, London, UK
[3]Department of Physics, University of Otago, Dunedin, New Zealand

**Correspondence:** Ville Maliniemi (ville.maliniemi@uib.no)

**Abstract.** Chlorofluorocarbon (CFC) emissions in the latter part of the 20th century reduced stratospheric ozone abundance substantially, especially in the Antarctic region. Simultaneously, polar stratospheric ozone is also destroyed catalytically by nitrogen oxides ($NO_x$=NO+$NO_2$) descending from the mesosphere and the lower thermosphere during winter. These are produced by energetic particle precipitation (EPP) linked to solar activity and space weather. Active chlorine ($ClO_x$=Cl+ClO) can also react mutually with EPP produced $NO_x$ or hydrogen oxides ($HO_x$) and transform both reactive agents into reservoir gases chlorine nitrate or hydrogen chloride, which buffer ozone destruction by all these agents. We study the interaction between EPP produced $NO_x$, ClO and ozone over the 20th century by using free running climate simulations of the chemistry-climate model SOCOL3-MPIOM. Substantial increase of $NO_x$ descending to polar stratosphere is found during winter, which causes ozone depletion in the upper and mid-stratosphere. However, in the Antarctic mid-stratosphere the EPP induced ozone depletion becomes less efficient after 1960s, especially during springtime. Simultaneously, significant decrease in stratospheric ClO and increase in hydrogen chloride and partly chlorine nitrate between 10-30 hPa can be ascribed to EPP forcing. Hence, interaction between EPP produced $NO_x$/$HO_x$ and ClO likely suppressed the ozone depletion due to both EPP and ClO at these altitudes. Furthermore, at the end of the century significant ClO increase and ozone decrease is obtained at 100 hPa altitude during winter and spring. This lower stratosphere response shows that EPP can influence activation of chlorine from reservoir gases on polar stratospheric clouds, thus modulating chemical processes important for ozone hole formation. Our results show that EPP has been a significant modulator of reactive chlorine in the Antarctic stratosphere during the CFC era. With the implementation of the Montreal Protocol, stratospheric chlorine is estimated to return to pre-CFC era levels after 2050. Thus, we expect increased efficiency of chemical ozone destruction by EPP-$NO_x$ in the Antarctic upper and mid-stratosphere over coming decades. The future lower stratosphere ozone response by EPP is more uncertain.

## 1 Introduction

Chlorofluorocarbon (CFC) emissions caused stratospheric ozone to decrease substantially during the latter half of the 20th century (WMO, 2018). This was especially dramatic in the Antarctic where the ozone hole formed in the lower stratosphere (Anderson et al., 1991).

Atmospheric chlorine released from the CFC emissions can destroy ozone via catalytic reactions like

$$ClO + O \rightarrow Cl + O_2. \tag{R1}$$

$$Cl + O_3 \rightarrow ClO + O_2. \tag{R2}$$

This reaction chain has a peak effectiveness in the upper stratosphere between 40 and 50km altitudes (Lary, 1997). For the most part of the year, chlorine in the lower stratosphere is stored in a reservoir gases like chlorine nitrate ($ClONO_2$) and hydrogen chloride (HCl) (Molina et al., 1987). During winter, the Antarctic lower stratosphere has cold enough temperatures to form polar stratospheric clouds (PSC) (Pitts et al., 2018). Heterogeneous reactions on PSCs break chlorine reservoir gases to reactive chlorine (Molina et al., 1987), leading to catalytic ozone depletion during spring via the chain of reactions:

$$ClO + ClO + M \rightarrow Cl_2O_2 \tag{R3}$$

$$Cl_2O_2 + h\nu \rightarrow Cl + ClOO \rightarrow 2Cl + O_2 \tag{R4}$$

$$2 \times (Cl + O_3 \rightarrow ClO + O_2) \tag{R5}$$

This reaction chain has a peak effectiveness at 15-20km altitude (Lary, 1997), and is important for the ozone hole formation. One can see the monthly Antarctic ozone climatology and the trend in the latter half of the 20th century in Figure 1.

Polar ozone can also be destroyed catalytically by reactive nitrogen oxides ($NO_x=NO+NO_2$) via reactions:

$$NO_2 + O \rightarrow NO + O_2 \tag{R6}$$

$$NO + O_3 \rightarrow NO_2 + O_2. \tag{R7}$$

This reaction chain peaks at 45km altitude (Lary, 1997). One of the main sources of polar $NO_x$ is energetic electron precipitation (EEP) from the magnetosphere, which ionizes the polar thermosphere and the upper mesosphere (Mironova et al., 2015; Nesse Tyssøy et al., 2019). In addition, sporadic solar proton events (SPE) can produce $NO_x$ in the mesosphere/upper stratosphere (Jackman et al., 2009). During winter, polar regions remain in darkness which prolongs the chemical lifetime of $NO_x$ (Solomon et al., 1982; Funke et al., 2014). Furthermore, downward vertical residual circulation in the wintertime transports polar $NO_x$ to stratospheric altitudes (Seppälä et al., 2007; Maliniemi et al., 2020). It has been shown that this indirect stratospheric $NO_x$ can deplete ozone by 10-15 percent in the Antarctic upper stratosphere during winter (Semeniuk et al., 2011; Damiani

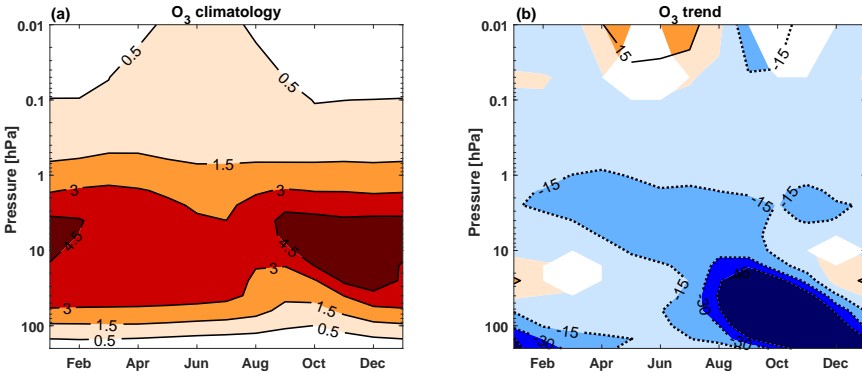

**Figure 1. a)** Zonal mean ozone climatology in the Antarctic (70°S-90°S, 0.01-200 hPa) during 1958-2008. Contour levels are 0.5, 1.5, 3 and 4.5 parts-per-million (ppm). **b)** Zonal mean ozone trend in the Antarctic (70°S-90°S, 0.01-200 hPa) during 1958-2008. Positive contour level (solid line) is 15 percent (%), and negative contour levels (dotted lines) are -15, -30 and -45 percent. Colour shading indicates areas significant at the 95% level calculated with a Mann-Kendall test and a false detection rate. Figures were made using the REF simulation (see details in Data and Methods).

et al., 2016; Arsenovic et al., 2019). Galactic cosmic rays (GCR) also impact atmospheric $NO_x$ and ozone. Their ionization peaks around 10-15 km affecting the upper troposphere/lower stratosphere region directly (Calisto et al., 2011; Jackman et al., 2016).

Reactive $NO_x$ gases also interact with stratospheric ClO via reactions

$$ClO + NO_2 + M \rightarrow ClONO_2 + M \tag{R8}$$

$$ClO + NO \rightarrow Cl + NO_2. \tag{R9}$$

Reaction R8 stores both reactive nitrogen and chlorine into a reservoir agent (Lary, 1997), and thus buffers the ozone destruction associated with both $NO_x$ and ClO. Reaction R9 couples catalytic ozone destruction cycles of chlorine and $NO_x$ (Brasseur and

Solomon, 2005). Formation of hydrogen chloride (HCl) via reactions with $HO_x$ can also be important for EPP impact:

$$ClO + OH \rightarrow HCl + O_2 \tag{R10}$$

$$ClO + OH \rightarrow Cl + HO_2 \tag{R11}$$

$$Cl + HO_2 \rightarrow HCl + O_2. \tag{R12}$$

EPP is known to produce hydrogen oxides ($HO_x$) in the mesosphere (Verronen et al., 2011), and direct SPE production can reach the upper stratosphere (Jackman et al., 2009). While $HO_x$ lifetime is too short for any EPP indirect stratospheric $HO_x$, formation of $HNO_3$ in reaction between $NO_2$ and OH, and its subsequent descent to stratospheric altitudes during polar night might also play a role (Verronen and Lehmann, 2015). Finally, reaction

$Cl + CH_4 \rightarrow HCl + CH_3$                                                                   (R13)

can also be important following the reaction (R9) (Brasseur and Solomon, 2005).

A recent study showed that satellite observations of Antarctic ClO correlated negatively with the geomagnetic activity during springtime (Gordon et al., 2021). This implies that ozone depletion by polar $NO_x$ is modulated by chlorine loading, especially during the CFC era. Similarly, stratospheric indirect $NO_x$ will modulate the ozone depletion by ClO. In this paper we investigate

how the interaction between $NO_x$, ClO and ozone and the emergence of the CFC era modulate the particle precipitation impact on ozone by using a free running chemistry-climate simulation with implemented particle precipitation forcing over the whole 20th century.

## 2   Data and Methods

The chemistry-climate model SOCOL3-MPIOM (Stenke et al., 2013; Muthers et al., 2014) consists of three interactively

coupled components. The atmospheric component is the general atmospheric circulation model ECHAM5.4 (Roeckner, 2003), here used in configuration with T31 spectral horizontal truncation (approximately 3.75°x3.75°) and 30 vertical levels from the surface to 0.01 hPa (∼80 km). ECHAM5.4 is used in free-running mode with prescribed quasi-biannual oscillation (QBO) in the zonal wind, as the model cannot generate the QBO with the applied vertical resolution. The chemistry module MEZON (Egorova et al., 2003) computes the tendencies of 41 gas species, including 200 gas-phase, 16 heterogeneous and 35 photolytic

reactions. The oceanic component is MPIOM (Marsland et al., 2003), used in the nominal horizontal resolution of 3° with 40 vertical layers from the ocean surface to the bottom.

The solar radiation input is based on the study by Shapiro et al. (2011). The precipitating energetic particles are prescribed following the Coupled Model Intercomparison Project Phase 6 (CMIP6) recommendations (Matthes et al., 2017). Medium-energy electrons (>30keV) are implemented as daily ionisation rates, while auroral electrons (<30keV) are represented as

NO influx through the model top (Funke et al., 2016). SPEs and GCR are also implemented as daily ionisation rates. The tropospheric aerosols originate from NCAR Community Atmospheric Model (CAM3.5) simulations with a bulk aerosol model forced with the Community Climate System Model 3 sea surface temperatures. The concentrations of greenhouse gases, ozone depleting substances and ozone precursors (CO and $NO_x$) follow historic values (Meinshausen et al., 2011).

In this study, the experiment simulation (EXP) contains all energetic particle precipitation sources (EEP+SPE+CGR) along

with all other forcings, while the reference simulation (REF) contains all forcings apart from the particle precipitation sources. Total simulation length is 109 years (1900-2008). Timeseries of geomagnetic activity (EEP ionization model is based on the Ap index (Matthes et al., 2017)), SPEs, and GCR can be seen in Figure 2. One can see clear increase in geomagnetic activity

from the early 1900s to 1960s, in agreement with grand solar maximum. This is also seen in the cosmic ray ionization, which reach centennial minimum at the same time (stronger heliospheric magnetic field is able to shield cosmic rays more efficiently). Frequency of the SPEs does not have any clear trend. One must note though that the SPE and GCR ionization before 1960s is only estimated indirectly based on the overall solar activity (Matthes et al., 2017). An eleven-member ensemble was generated by varying initial $CO_2$ concentrations with 0.1% among the different members of EXP and REF.

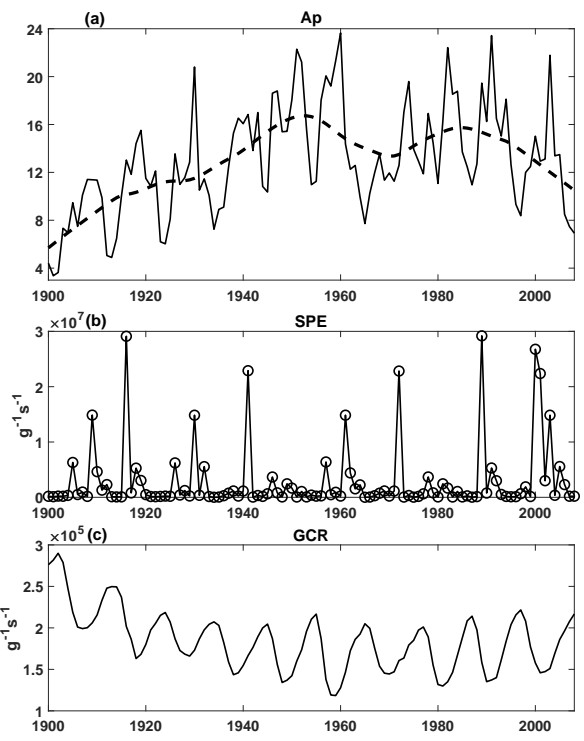

**Figure 2. a)** Annual time series of geomagnetic activity index Ap (solid line) and 31-year smooth trend (dotted line). **b)** Annual time series of SPEs (ion pair production rate at 1 hPa, 70°S-90°S). **c)** Annual time series of GCR (ion pair production rate at 100 hPa, 70°S-90°S).

For the atmospheric parameters we compute zonal averages and obtain their monthly latitude-height profiles and analyse the EPP response (EXP-REF). We concentrate on volume mixing ratios of $NO_x$, ClO, $ClONO_2$, HCl and ozone. Significance in each lat-height bin is calculated with a monte carlo simulation. We combine ensembles from both EXP and REF (22 total) and randomly take two 11-ensemble data collections 10,000 times. These two randomly picked data matrices are subtracted similarly as the original data in each lat-height grid. Original value (EXP-REF) is compared to the distribution of these 10,000 repetitions to obtain the fraction of more extreme differences (both tails of the distribution). This fraction then represents the p-value in each lat-height bin with the null hypothesis that there is no difference between EXP and REF.

Results presented in a latitude-height grid are usually spatially correlated, and represent a multiple hypothesis testing situation (Wilks, 2016). Thus, simply presenting significance in each bin based on individual hypothesis testing will lead to an

overestimation of the true number of rejected null-hypotheses. The average number of false positives is n*p, where n is the number of hypothesis tests and p is the used p-value. This is due to the definition of the p-value, and also the dependency of the neighbouring grid points, i.e., the spatial autocorrelation (Wilks, 2016). To overcome this issue, the false discovery rate is calculated for each case. It reduces the probability of false positives in line with the applied new p-value limit. After the procedure, the p-value can be interpreted as being the probability to obtain a false rejection of a null-hypothesis. Details of the method can be found in Wilks (2016). We have used the p-value limit of 0.05, which then represents 95% probability of not having any false positives.

Figure 1 shows the relative change in ozone over the Antarctic from 1958 to 2008 in the REF simulation calculated by subtracting a 5-year mean centered on 1960 from a 5-year mean centered on 2006. Significance is calculated using a Mann-Kendall test (Mann, 1945) and a false discovery rate. The smooth long-term variations shown in Figure 2 and the following Figures 7-10 are calculated using the LOWESS-method (LOcally WEighted Scatterplot Smoothing) applied with a 31-year window (Cleveland and Devlin, 1988). More details of the method can be found in Maliniemi et al. (2014).

## 3 Results

### 3.1 Global $NO_x$, ClO and $O_3$ responses to EPP during 1979-2008

Figure 3 shows monthly $NO_x$ volume mixing ratio increase in the mesosphere and the stratosphere due to the EPP forcing during 1979-2008. $NO_x$ reaches stratospheric altitudes (below 1 hPa) in the polar regions during local winter. This is mainly due to EEP and to a lesser extent caused by SPEs (Maliniemi et al., 2020), e.g., there were only 3-4 notable SPEs during 1979-2008 as seen in Figure 2b. In the Antarctic stratosphere substantial increase of $NO_x$ is obtained from May until February, reaching roughly 20-30 hPa by late winter/spring. In the Arctic stratosphere the lowest altitude is slightly higher, around 10 hPa during March.

EPP leads to ozone depletion in the polar regions in the mesosphere and upper stratosphere as shown in Figure 4. Mesospheric ozone depletion is dominated by $HO_x$ production (Jackman et al., 2008; Andersson et al., 2014; Zawedde et al., 2019). However, mesospheric ozone depletion seems to be slightly stronger in the southern hemisphere than in the northern hemisphere, which implies an additional dynamical or long-lived component in the southern hemisphere. This can be explained by southern hemispheric polar vortex forming earlier and being more stable and less diffusive, which confines ozone depleted air more efficiently (Andersson et al., 2018). Additional explanation can also be $HNO_3$ formation in the mesosphere (Verronen and Lehmann, 2015). Thermospheric NO in the model is prescribed with a semi-empirical model, which on average has more NO entering the mesosphere in the southern hemisphere (Funke et al., 2016).

The stratospheric ozone depletion is due to $NO_x$ catalytic reactions (R6-R7) (Lary, 1997). It is larger in magnitude and lasts notably longer in the southern hemisphere than in the northern hemisphere, because the strong and stable polar vortex in the southern hemisphere isolates air mass efficiently. Seasonal evolution of the EPP ozone impact is in good agreement with earlier studies (Rozanov et al., 2012; Damiani et al., 2016). There is also a weak but significant ozone response around 100 hPa altitude covering all latitudes. It is positive and significant in the low latitudes all year and in the high latitudes during

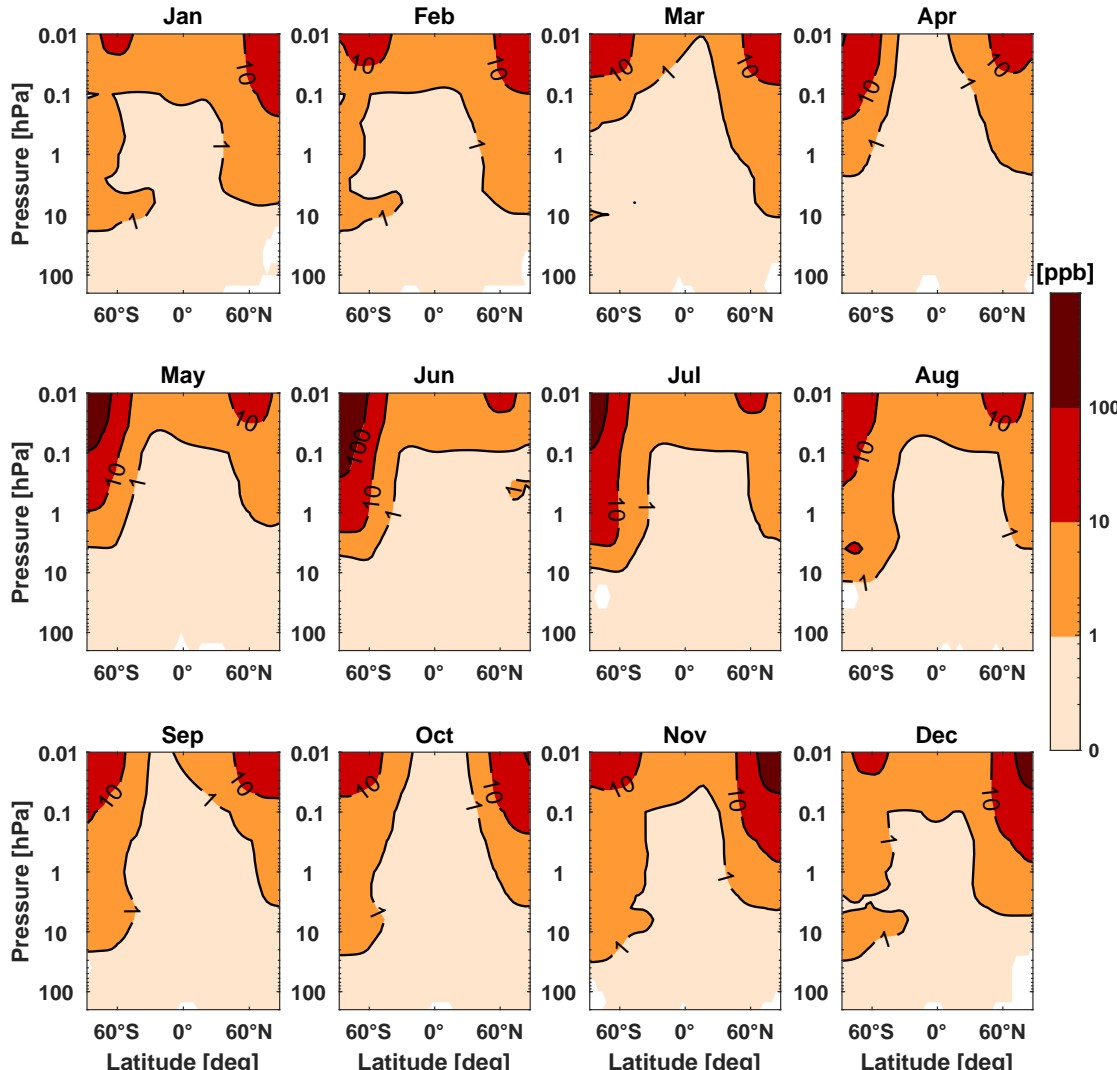

**Figure 3.** Absolute difference in the monthly zonal mean $NO_x$ between EXP and REF during 1979-2008. Positive contour levels are 1, 10 and 100 parts-per-billion (ppb). Altitude range is from 0.01 hPa to 200 hPa. Colour shading indicates areas significant at the 95% level calculated with a monte carlo simulation and a false detection rate.

summer, but significantly negative during winter/spring, at least in the Antarctic. This Antarctic lower stratosphere response is also in agreement with Rozanov et al. (2012) and Damiani et al. (2016). Consistently positive weak ozone response in the lower stratosphere is a consequence of GCR (Calisto et al., 2011; Jackman et al., 2016), while it is likely dominated by the indirect EPP effect in the high latitudes during winter.

Figure 5 shows monthly ClO volume mixing ratio related to the EPP forcing. Altitude of polar ClO decrease in the southern hemisphere agrees well with the lowest altitude of $NO_x$ increase during May-November. There is a substantial decrease of

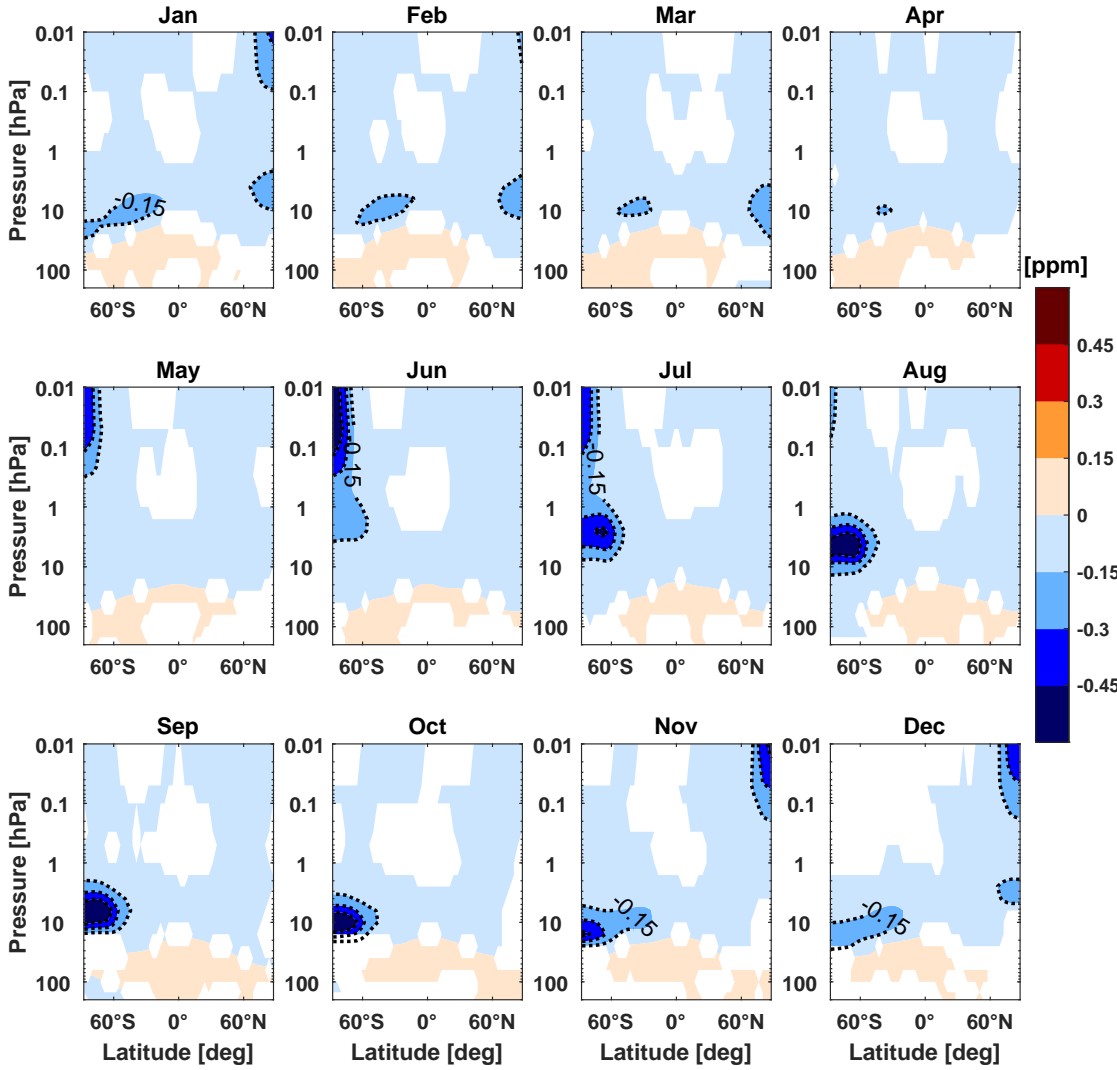

**Figure 4.** Absolute difference in the monthly zonal mean O$_3$ between EXP and REF during 1979-2008. Negative contour levels are -0.15, -0.3 and -0.45 parts-per-million (ppm). Colour shading indicates areas significant at the 95% level calculated with a monte carlo simulation and a false detection rate.

ClO below 1 hPa from June to November in the Antarctic stratosphere, which extends down to 30 hPa in springtime matching the pattern of descending NO$_x$. As explained above, NO$_x$ and ClO can interact with reactions R8 and R9 (HO$_x$ and ClO via reactions R10-R12), interfering with the catalytic ozone destroying cycles of both agents. Interestingly, there are weak but significantly positive ClO responses in the Antarctic at 100 hPa altitude during winter (June to September).

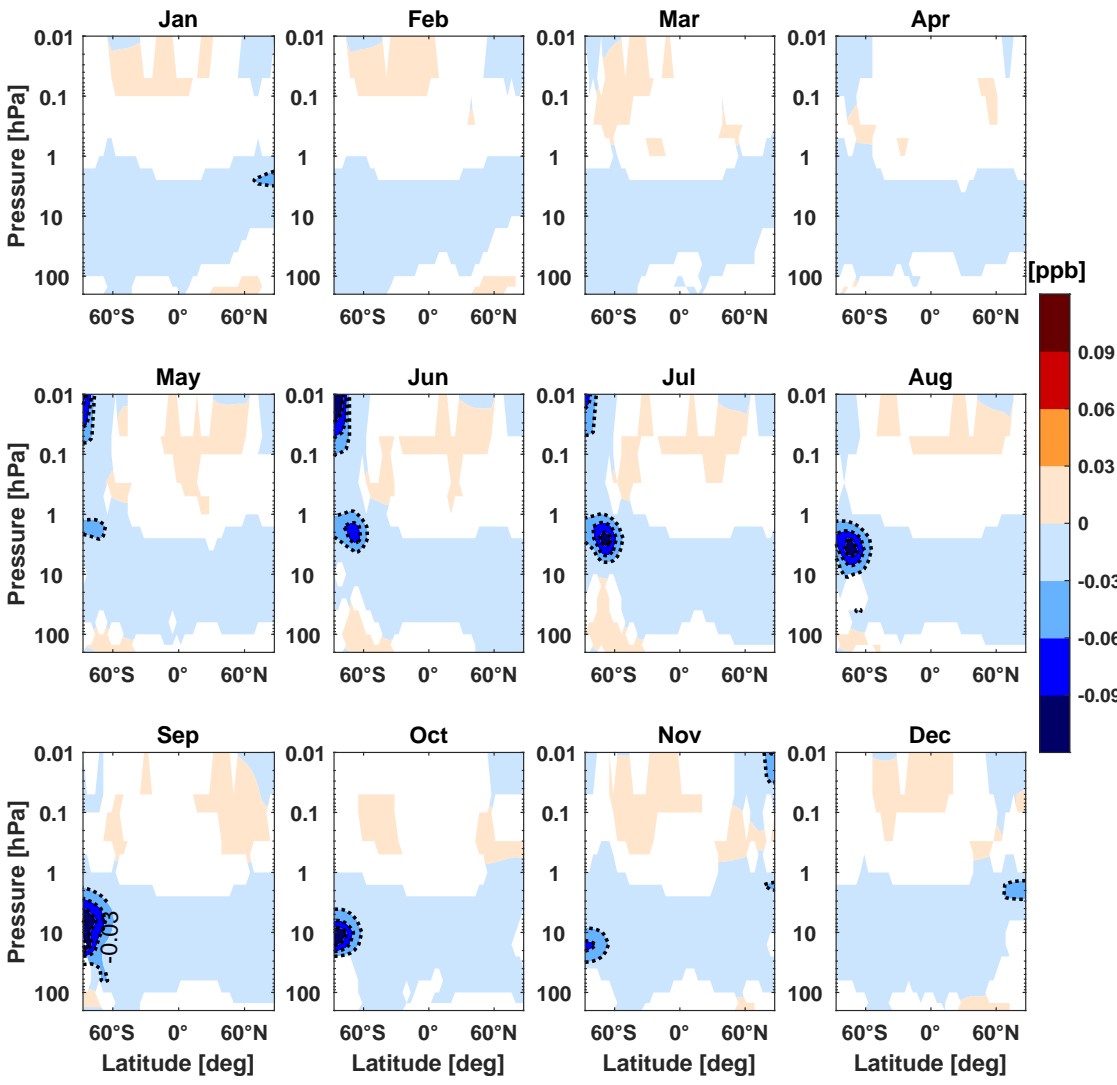

**Figure 5.** Absolute difference in the monthly zonal mean ClO between EXP and REF during 1979-2008. Negative contour levels are -0.03, -0.06 and -0.09 ppb. Colour shading indicates areas significant at the 95% level calculated with a monte carlo simulation and a false detection rate.

## 3.2 Antarctic $NO_x$, ClO and $O_3$ responses to EPP over the 20th century

Figure 6 shows the Antarctic polar climatologies of $NO_x$, ClO, $ClONO_2$ and HCl during 1979-2008 from the REF simulation. Small values are denoted in all variables below 200 hPa (not shown). Furthermore, ClO and $ClONO_2$ have small values above the stratopause (roughly 1 hPa).

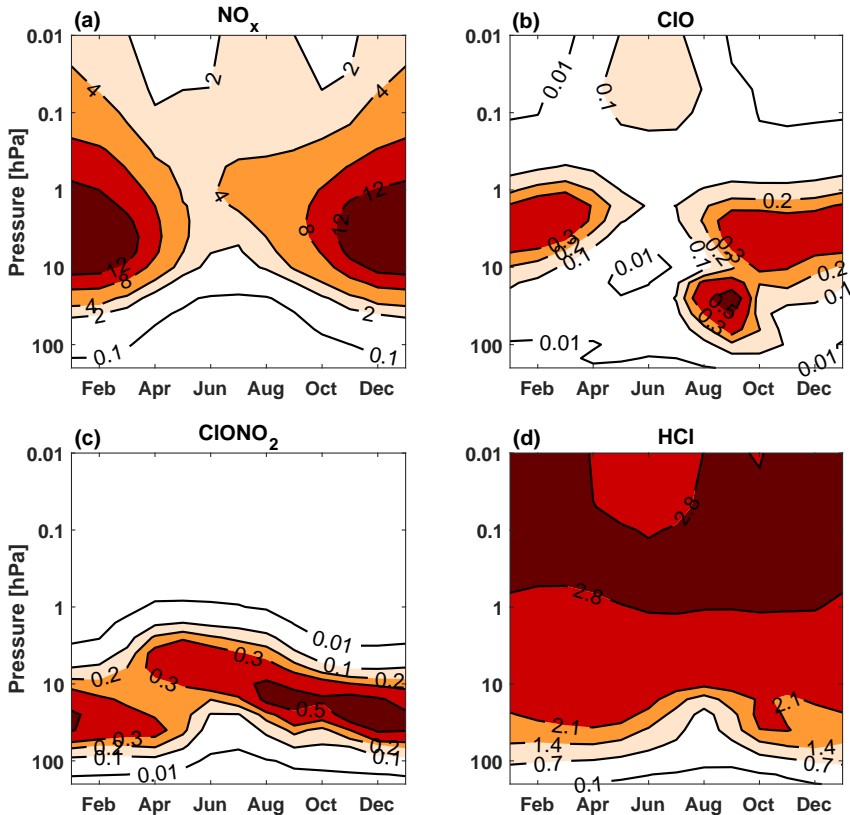

**Figure 6. a)** Seasonal climatology of $NO_x$ in the Antarctic (70°S-90°S, 0.01-200 hPa) during 1979-2008. Positive contour levels are 0.1, 2, 4, 8 and 12 ppb. Seasonal climatology for **b)** ClO and **c)** $ClONO_2$ are shown with positive contour levels of 0.01, 0.1, 0.2, 0.3 and 0.5 ppb, and for **d)** HCl with positive contour levels of 0.1, 0.7, 1.4, 2.1 and 2.8 ppb. All figures are calculated using the REF simulation, i.e., without the EPP forcing.

Seasonally varying $NO_x$ increase in the Antarctic stratosphere due to EPP is shown in Figure 7a. In addition, September-October $NO_x$ abundance at 10-20 hPa in EXP and REF are compared over the whole 20th century in Figures 7b and 7c. $NO_x$ increase of more than a hundred percent is present in the upper stratosphere during winter. $NO_x$ descends below 10 hPa by July, and more than 50 percent increase is sustained until November when the polar vortex typically breaks down. There is also a relative increase of $NO_x$ around 50-100 hPa during June-September. This is directly related to GCR and partly to EEP/SPE from previous season (more than 15 percent increase continues descending after November). Similar increase at these altitudes during mid-winter is also obtained in Rozanov et al. (2012).

The 10-20 hPa $NO_x$ timeseries for September-October shows that there is a consistent negative trend over the whole 20th century in REF, possibly a consequence of increasing chemical destruction of $NO_x$ in the cooler stratosphere (Stolarski et al., 2015). The $NO_x$ timeseries in EXP is mostly affected by the EEP/geomagnetic activity (Figure 2a). Spearman rank correlation

coefficient between EXP NO$_x$ in Figure 7b and annual Ap index is 0.73 (p-value<0.01). Spearman correlation between EXP-REF NO$_x$ in Figure 7c and annual Ap index is 0.84. However, there is also a negative trend in the EXP NO$_x$ timeseries after 1960 in Figure 7b. The EXP NO$_x$ level at the end of the simulation is lower than in the beginning despite higher geomagnetic activity in the early 21st century compared to the early 20th century.

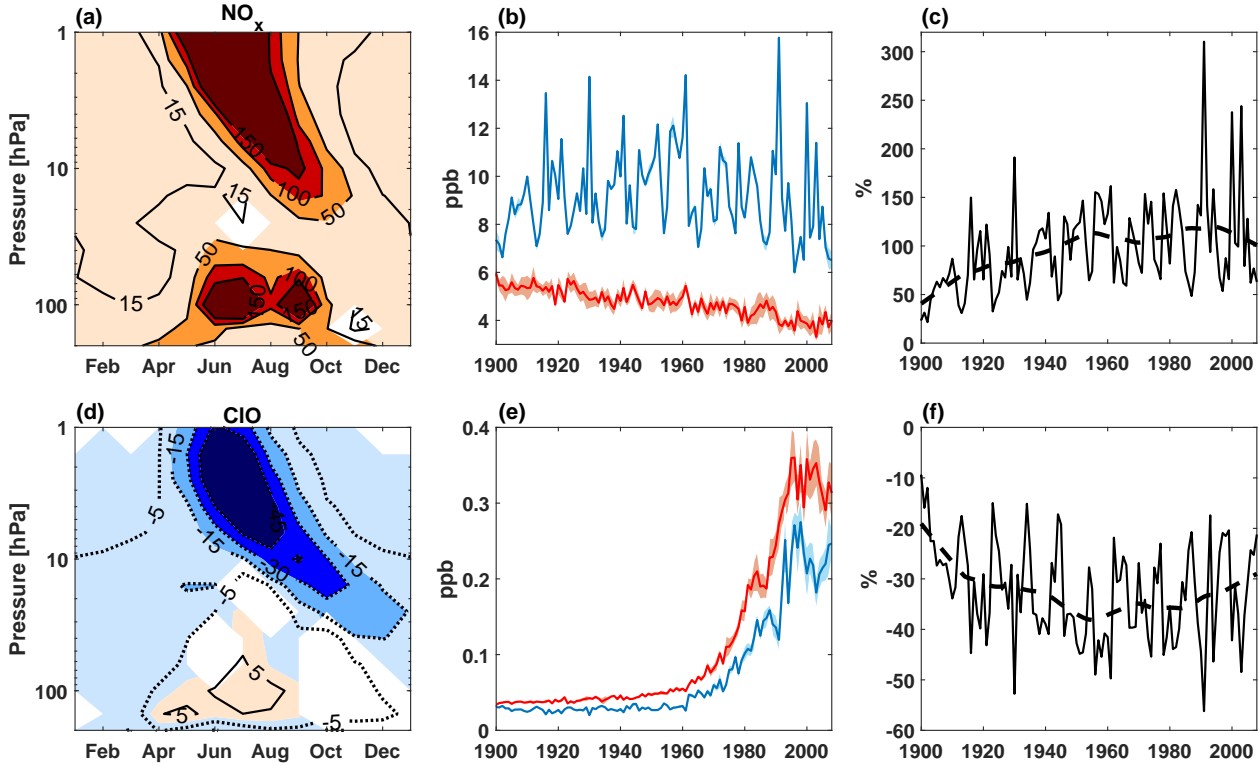

**Figure 7. a)** Relative difference (EXP-REF)/REF in the monthly zonal mean polar NO$_x$ (70°S-90°S, 1-200 hPa) during 1979-2008. Positive contour levels are 15, 50, 100 and 150 percent (%). Colour shading indicates areas significant at the 95% level calculated with a monte carlo simulation and a false detection rate. **b)** September-October ensemble mean time series of polar NO$_x$ (70°S-90°S, 10-20hPa) in EXP (blue) and REF (red). Shaded light blue and red areas represent 95% confidence intervals of ensemble means. **c)** Relative difference of EXP and REF (solid line) and 31-year smooth trend (dotted line). **d)** Polar ClO with negative contour levels of -5, -15, -30, -45 percent (dotted lines) and positive contour level of 5 percent (solid line). **e)** and **f)** Same for ClO as **b)** and **c)**, respectively.

Figure 7d shows relative changes in Antarctic stratospheric ClO due to the EPP forcing. More than 45 percent decrease in ClO is obtained between EXP and REF in mid-winter between 1 and 10 hPa in Figure 7d. More than 15 percent ClO decrease in EXP relative to REF continues well into spring extending down to 40 hPa.

The mid-stratospheric ClO timeseries in September-October (Figures 7e and 7f) shows that there is a strong increase in chlorine since 1960, due to CFC emissions. However, the ClO amount in EXP consistently falls below the amount in REF. The relative difference between EXP and REF is anticorrelating with the level of geomagnetic activity (R=-0.46, p-value<0.01)

rather than being dependent on the overall amount of ClO. ClO is reduced by 30-40% with the EPP forcing at 10-20 hPa in September-October after the 1950s.

Interestingly, significant positive ClO response to EPP forcing at 50-100 hPa is present during winter months in Figure 7d. This occurs at altitudes where polar stratospheric clouds (PSC) form in the Antarctic (Pitts et al., 2018), likely being a consequence of reservoir gases breaking on PSC (Molina et al., 1987; Webster et al., 1993). Influence of EPP on these processes in the lower stratosphere is discussed in more detail below. This positive ClO response in the lower stratosphere was also seen in satellite data by Gordon et al. (2021) during August-October.

Figure 8 shows the relative changes in Antarctic stratospheric $ClONO_2$ and $HCl$ due to the EPP forcing. Substantial increase of $ClONO_2$ in EXP relative to REF is seen in the upper stratosphere during winter (Figure 8a). However, if compared to the climatology of $ClONO_2$ in Figure 6, it is evident that at altitudes above 3 hPa there is very little $ClONO_2$ and most is located between 3 hPa and 80 hPa. Between 3 and 10 hPa, $ClONO_2$ amount is less in EXP than in REF during mid-winter. This implies that reduced ClO due to EPP at this location in Figure 7d cannot be explained by reaction R8, but is likely due to reaction R9 and consequent formation of $HCl$ seen in Figure 8d (Cl reacting with methane) (Brasseur and Solomon, 2005). Reactions (R10-R12) with SPE produced OH, and/or formation of mesospheric $HNO_3$ (Verronen and Lehmann, 2015) and its subsequent descent to stratospheric altitudes during polar night with following photolysis might partly play a role in EPP related $HCl$ formation.

In spring time there is a significant $ClONO_2$ increase due to EPP at 10-20 hPa altitudes (Figure 8a), which is also seen in timeseries Figures 8b and 8c. The relative effect of EPP on $ClONO_2$ increases after 1960, reaching 5% after 2000. However, positive correlation between annual geomagnetic activity and $HCl$ in 10-20 hPa from Figure 8f (R=0.48, p-valu<0.01) implies that ClO is converted to $HCl$, instead of $ClONO_2$ (correlation between geomagnetic activity and Figure 8c data is -0.01 over the whole time period).

There is also a significant and strong increase of $ClONO_2$ in the lower stratosphere during winter in Figure 8a accompanied with a significant decrease of $HCl$ in Figure 8d. It implies that EPP is able to interact with the mechanism of ClO release from $ClONO_2$ and $HCl$ on PSC (Molina et al., 1987; Webster et al., 1993). ClO is mostly released by consuming $HCl$ which recovers slowly to initial values, while $ClONO_2$ levels can rebuild fairly quickly and to excess levels of its initial values (Molina et al., 1987; Webster et al., 1993), especially in the presence of additional $NO_x$ due to EPP. This means that EPP keeps the heterogeneous processing between $HCl$ and $ClONO_2$ running and results in less $HCl$, more $ClONO_2$ and finally more ClO via processing on PSCs.

The relative effect of EPP on Antarctic ozone shows 10-15 percent decrease in the upper and mid-stratosphere from mid-winter until November (Figure 9a). Timeseries of mid-stratospheric (10-20 hPa) ozone in both EXP and REF during September-October (Figure 9b) show a negative trend since 1960, but it is notably weaker in EXP. The relative difference between EXP and REF (Figure 9c) shows that the trend correlates negatively with the overall geomagnetic activity before the CFC era (Spearman correlation R=-0.65, p-value<0.01 during 1900-1960). After 1960 the effect of EPP on ozone at 10-20 hPa diverges away from the overall geomagnetic activity level (R=-0.07, p-value=0.65 during 1961-2008). At the beginning of the 21st century, average

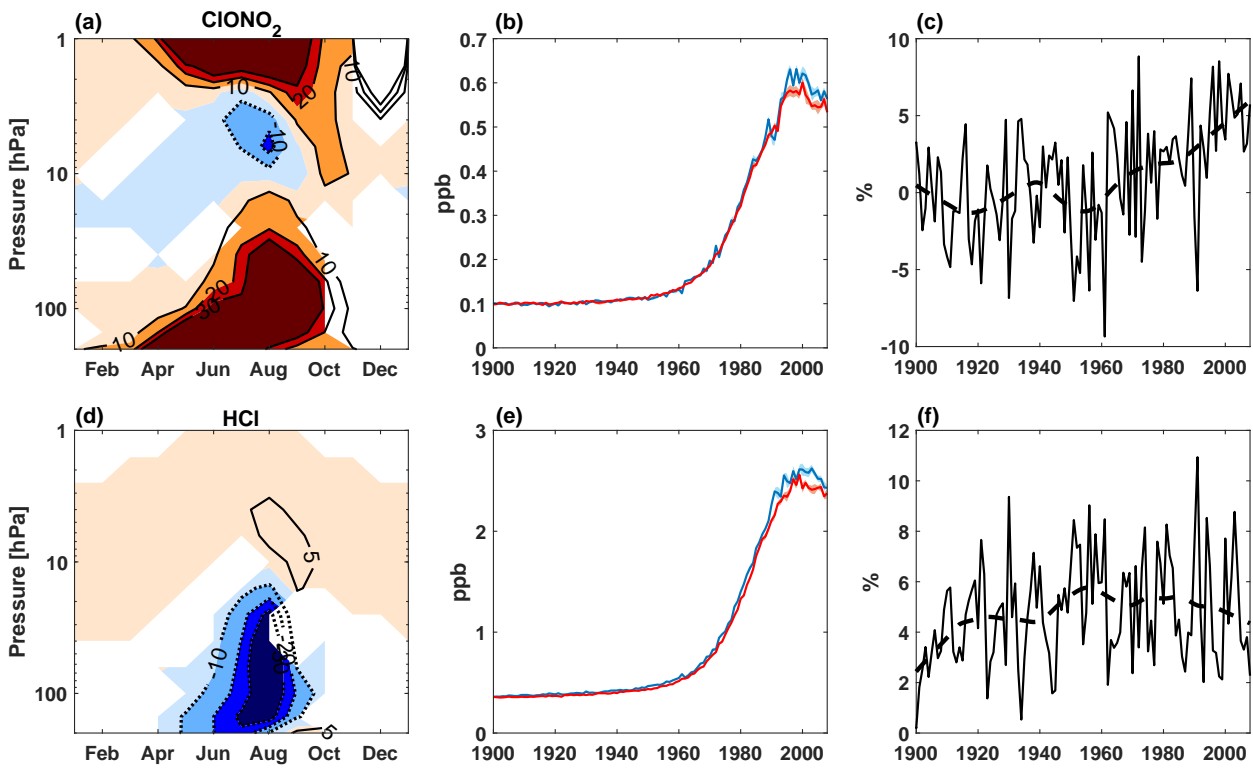

**Figure 8. a)** Relative difference (EXP-REF)/REF in the monthly zonal mean polar ClONO$_2$ (70°S-90°S, 1-200 hPa) during 1979-2008. Positive contour levels (solid lines) are 10, 20 and 30 percent (%) and negative contour level (dotted line) is -10 percent. Colour shading indicates areas significant at the 95% level calculated with a monte carlo simulation and a false detection rate. **b)** September-October ensemble mean time series of polar ClONO$_2$ (70°S-90°S, 10-20hPa) in EXP (blue) and REF (red). Shaded light blue and red areas represent 95% confidence intervals of ensemble means. **c)** Relative difference of EXP and REF (solid line) and 31-year smooth trend (dotted line). **d)** Polar HCl with negative contour levels of -10, -20 and -30 percent (dotted lines) and positive contour level of 5 percent (solid line). **e)** and **f)** Same for HCl as **b)** and **c)**, respectively.

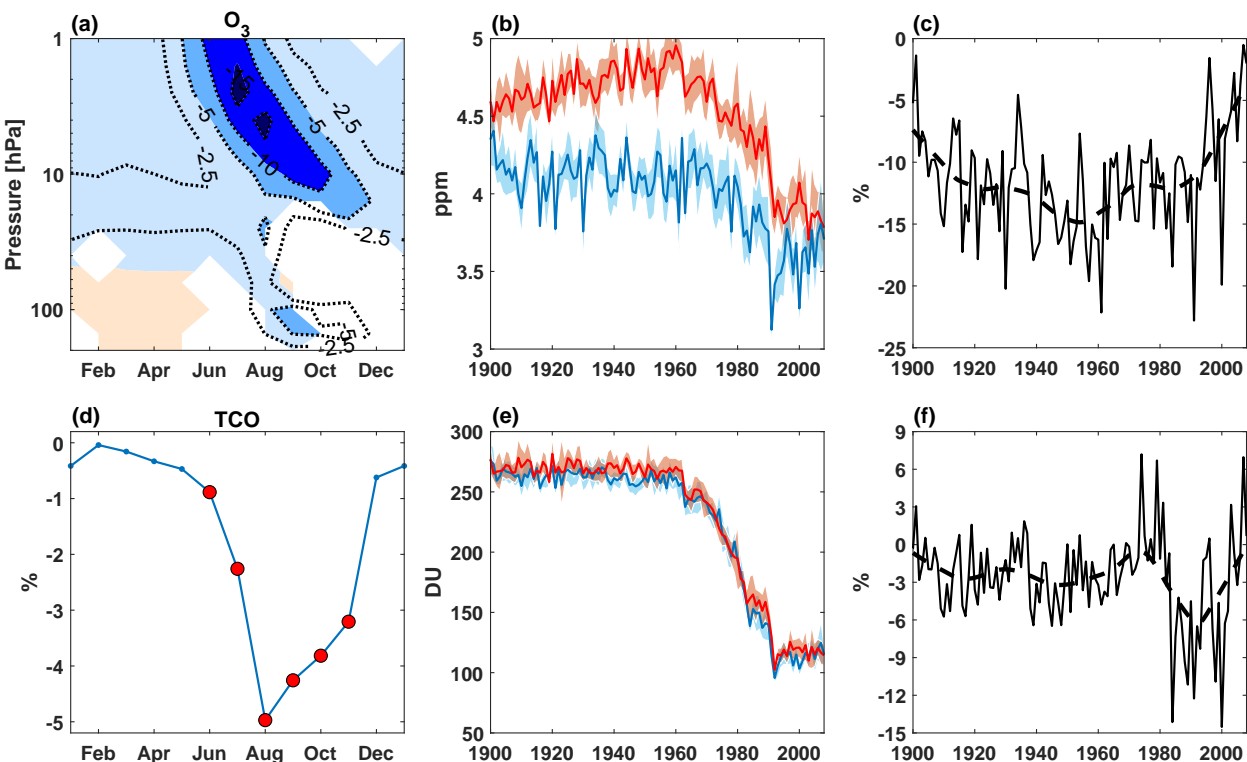

**Figure 9. a)** Relative difference (EXP-REF)/REF in the monthly zonal mean polar ozone (70°S-90°S, 1-200 hPa) during 1979-2008. Negative contour levels (dotted lines) are -2.5, -5, -10 and -15 percent (%). Colour shading indicates areas significant at the 95% level calculated with a monte carlo simulation and a false detection rate. **b)** September-October ensemble mean time series of polar ozone (70°S-90°S, 10-20hPa) in EXP (blue) and REF (red). Shaded light blue and red areas represent 95% confidence intervals of ensemble means. **c)** Relative difference of EXP and REF (solid line) and 31-year smooth trend (dotted line). **d)** Relative effect of EPP on polar total column ozone (TCO). Red circles represent months with significant difference at the 95% level calculated with a monte carlo simulation and a false detection rate. **e)** and **f)** Same for TCO as **b)** and **c)**, respectively.

ozone depletion due to EPP at 10-20 hPa is just a few percents, and is notably less than in the early 20th century (10 to 15%) while geomagnetic activity in the early 21st century is roughly at the level of 1920s/1930s level (Figure 2).

Figure 9a also shows a significant ozone depletion by EPP around 100 hPa altitude during August-October. This is in agreement with ClO increase seen in Figure 7d at same altitude earlier in winter. When polar night ends, ClO released by PSCs is depleting ozone via the ClO dimer cycle (R3-R5) (Lary, 1997).

The relative effect of EPP on Antarctic total column ozone (TCO) during 1979-2008 is shown in Figure 9d. It shows 4 to 5 percent decrease in spring (Aug-Oct). When considering the effect on TCO over the whole century in September-October

(Figure 9f), it has an average of around 1-3% reduction until 1980s after which it becomes notably stronger. To understand this evolution in TCO, Figure 10 shows 1900-2008 timeseries of August-October ozone at 100 hPa. Relative ozone effect at 100

hPa by EPP is mostly positive (Calisto et al., 2011) until 1980 after which it decreases to minus 5-10 percent level in Figure 10d. This is very different evolution than in the mid-stratosphere in Figure 9c. On the other hand, the EPP effect on TCO after 2000 returns to similar levels than before 1980, and seems to be above zero at the end of the simulation. Gordon et al.

(2021) showed positive correlation between geomagnetic activity and polar TCO in springtime (Oct-Nov) during 2005-2017. Spearman correlation between geomagnetic activity and EXP polar TCO during Oct-Nov 1998-2008 is also slightly positive but insignificant (R=0.21, p-value=0.54). This positive TCO response is likely due to reduction in mid-stratospheric ozone depletion as seen in Figure 9c (note roughly an order of magnitude difference in ozone abundance between 10-20 hPa and 100 hPa in Figures 9b and 10c).

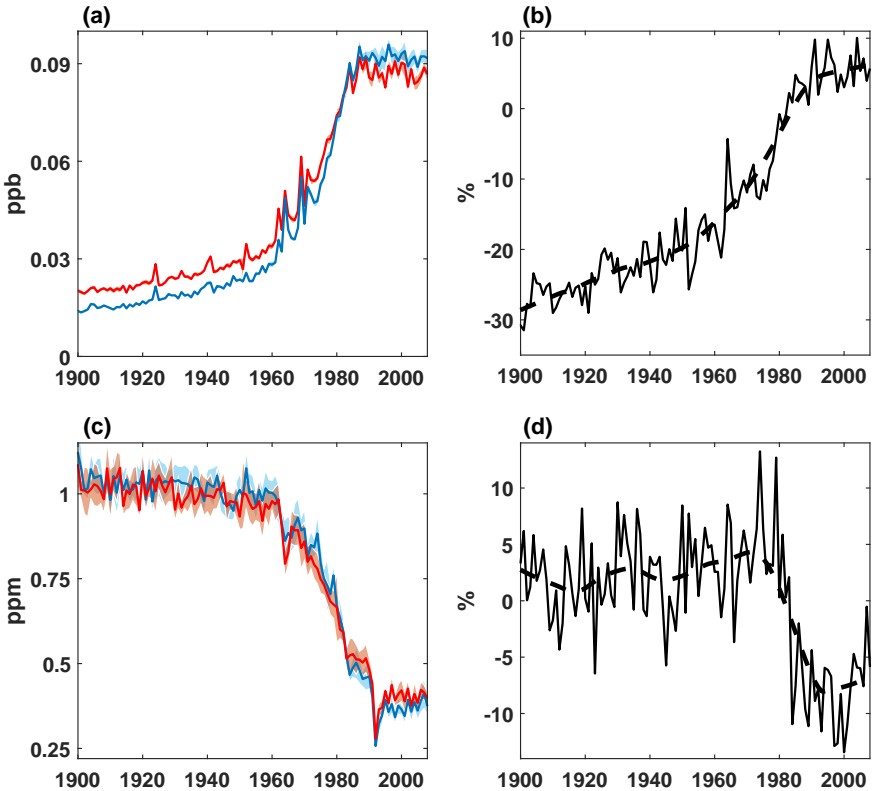

**Figure 10. a)** Jul-Sep ensemble mean time series of polar ClO (70°S-90°S, 100hPa) in EXP (blue) and REF (red). Shaded light blue and red areas represent 95% confidence intervals of ensemble means. **b)** Relative difference of ClO EXP and REF (solid line) and 31-year smooth trend (dotted line). **c)** Aug-Oct ensemble mean time series of polar $O_3$ (70°S-90°S, 100hPa) in EXP (blue) and REF (red). Shaded light blue and red areas represent 95% confidence intervals of ensemble means. **d)** Relative difference of $O_3$ EXP and REF (solid line) and 31-year smooth trend (dotted line).

The absolute ClO timeseries at 100 hPa seen in Figure 10a increases substantially after 1960. Interestingly, the relative difference between EXP and REF (Figure 10b) follows the overall level of ClO (and CFC emissions), and is a very different than in the mid-stratosphere in Figure 7f.

    To understand this change of sign in lower stratospheric ClO response to EPP, we analyze seasonal variability of 100 hPa $NO/NO_2$ ratio, ClO, HCl and $ClONO_2$ responses between low and high chlorine loading in Figure 11. Time periods of 1924-

1934 (low chlorine) and 1997-2007 (high chlorine) were chosen since they represent roughly similar geomagnetic activity levels (see Figure 2a) during solar cycles 16 and 23, respectively. One can see that the $NO/NO_2$ ratio starting from June is notably higher during the CFC era (Figure 11b) than in the pre-CFC era (Figure 11a). Potential explanation for this is lower ozone amount which reduces reaction (R7) and increases the $NO/NO_2$ ratio. There is generally more $NO_x$ available (due to EPP) to react with $ClO_x$. In the pre-CFC era, low $NO/NO_2$ ratio favors $ClONO_2$ reformation (reaction R8) after heterogeneous

reaction between HCl and $ClONO_2$ (Figure 11c) resulting with less ClO and more $ClONO_2$ due to EPP (Figure 11e). Thus, heterogeneous processing on PSCs is HCl-limited in the pre-CFC era (Figure 11c). In the CFC era, higher $NO/NO_2$ ratio limits $ClONO_2$ reformation and allows active chlorine (and ClO) to accumulate (Figure 11d). Heterogeneous processing is thus $ClONO_2$-limited. While EPP related $ClONO_2$ production in the CFC era is also slowed due to higher $NO/NO_2$ ratio, it is still net positive over the whole winter (Figure 11f). This excess $ClONO_2$ due to EPP is able to keep the heterogeneous

processing running, resulting in ClO increase and considerable HCl decrease over the whole winter (Figure 11f).

    Positive ClO response in the lower stratosphere due to EPP was also found by Gordon et al. (2021). Damiani et al. (2016) showed negative ozone response at same altitudes related to EPP, albeit they were related to regression analysis with the geomagnetic activity. We note that it can be difficult to point to exact source (direct effect by GCR or indirect effect from above by EEP/SPE) in regression studies due to somewhat high collinearity between geomagnetic activity and GCR on interannual

scales (Maliniemi et al., 2019). Jackman et al. (2009) showed that nitrogen species ($NO_y$) and $ClONO_2$ produced by SPE survive long enough to descend to 100 hPa altitude several months after initial event. These results support that ClO response seen in the lower stratosphere is likely a combination of both EEP/SPE and GCR. We also note that this lower stratospheric ClO/ozone signal is not related to dynamics, i.e., any indirect effects of EPP on the polar vortex and meridional circulation. Any notable or significant zonal wind responses (EXP-REF) during winter 1979-2008 was not found (not shown).

**4 Summary**

This study verifies the significant polar stratospheric ozone depletion by EPP during winter over the whole 20th century. The ozone depletion in the upper and mid-stratosphere is a consequence of mostly EEP/geomagnetic activity and partly SPE producing $NO_x$ in the thermosphere and the mesosphere, which then descends to stratospheric altitudes during winter (Seppälä et al., 2007; Funke et al., 2014). In the Antarctic during 1979-2008, ozone depletion varies from more than 10% in the winter

upper stratosphere to 5-10% in the spring mid- and lower stratosphere between EXP and REF simulations. This is largely in agreement with previous studies (Rozanov et al., 2012; Damiani et al., 2016). The effect of EPP on Antarctic total column ozone is a few percent reduction during August to October, though it diminishes close to zero percent at the end of the simulation.

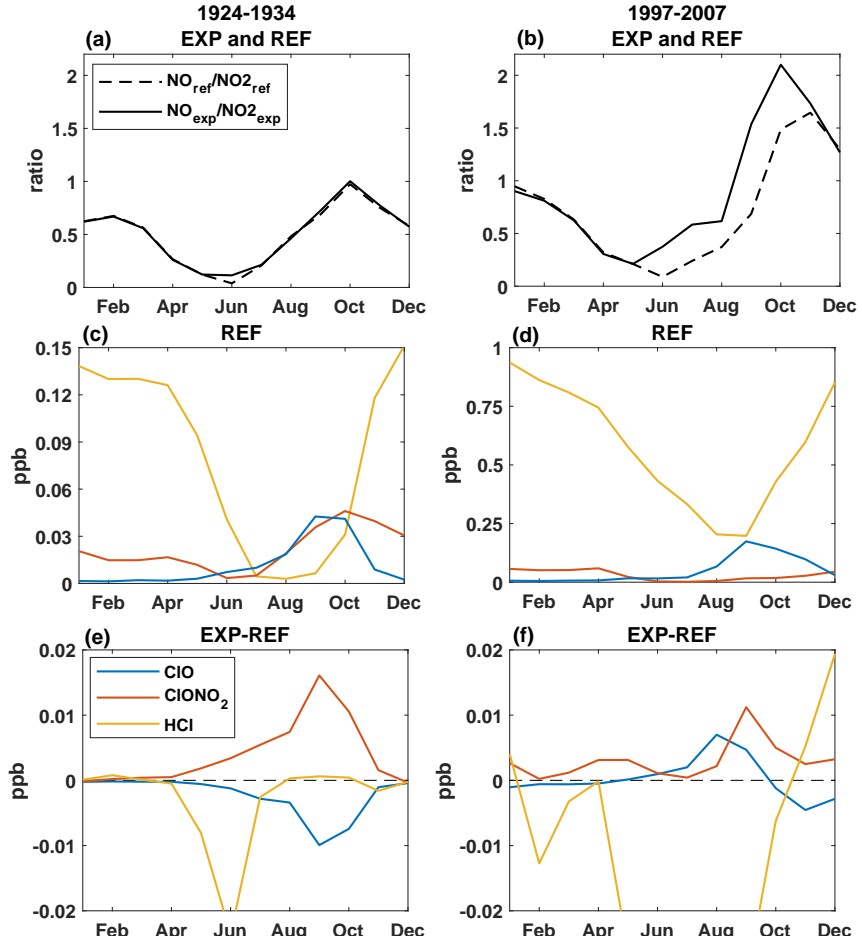

**Figure 11. a)** Seasonal polar NO/NO$_2$ ratio at 100 hPa in EXP (solid line) and REF (dashed line) during 1924-1934, and **b)** during 1997-2007. **c)** Seasonal REF polar 100 hPa climatology of ClO (blue), HCl (yellow) and ClONO$_2$ (red) during 1924-1934, and **d)** during 1997-2007. Note the different y-axes in **c)** and **d)**. **e)** Seasonal EXP-REF polar ClO (blue), HCl (yellow) and ClONO$_2$ (red) at 100 hPa during 1924-1934, and **f)** during 1997-2007.

However, the Antarctic ozone depletion is modulated during the latter half of the 20th century, especially during springtime. The ozone depletion efficiency in the mid-stratosphere weakens towards the beginning of the 21st century. Decadal geomagnetic activity decline in the late 20th century cannot solely account for this weaker ozone depletion by EPP. Furthermore, significant ozone depletion by EPP emerges during August-October in the lower Antarctic stratosphere (100 hPa) after 1980.

We find a significant decrease of stratospheric ClO due to the EPP impact in the same altitude where NO$_x$ is descending seasonally. The relative ClO reduction in the Antarctic upper and mid-stratosphere varies mainly with the level of geomagnetic activity over the whole century. EPP can reduce ClO by 30% in the upper stratosphere during winter and in the mid-stratosphere

during late winter/spring, even during the CFC era. ClO reduction during mid-winter between 1 and 20 hPa is mostly accounted by increase of HCl and partly by $ClONO_2$.

     In the lower Antarctic stratosphere (100 hPa), ClO abundance increases relative to the model run without EPP by more than 5 percent during winter after 1980, while EPP effect on ClO before the CFC era is negative. The seasonal emergence of this ClO response after 1980 is consistent with the formation of PSCs in the Antarctic and occurs slightly before the depletion of ozone

at the same altitude. Activation of chlorine from reservoir species $ClONO_2$ and HCl can be explained by heterogenous reactions on PSCs (Molina et al., 1987; Webster et al., 1993). We propose that during the pre-CFC era, heterogeneous processing on PSCs is HCl-limited and excess $NO_x$ due to EPP enhances $ClONO_2$ reformation via reaction (R8) leading to lower ClO levels. However, during the CFC-era low ozone levels limit reaction (R7) between NO and ozone and leads to higher $NO/NO_2$ ratio. This limits the $ClONO_2$ reformation and makes heterogeneous processing $ClONO_2$-limited. While EPP related production of

$ClONO_2$ is also slowed in the CFC era, it is still net positive over the whole winter. EPP is thus able to keep heterogeneous processing running and results in increase of ClO and substantial decrease of HCl. These results imply that EPP has significantly modulated chemical processes responsible for ozone hole formation.

     The introduction of CFC emissions since 1960s has significantly influenced the ozone response by EPP. With the implementation of the Montreal Protocol (Velders et al., 2007), ClO amount in the stratosphere is expected to return to pre-CFC

levels sometime after the 2050s. Based on results presented here, we can therefore expect higher efficiency of chemical ozone destruction by EPP in the upper stratosphere in the future. Furthermore, recent results have shown that EPP related $NO_x$ is increasing substantially in the future Antarctic upper stratosphere due to stronger vertical transport under climate change (Maliniemi et al., 2020, 2021). This is happening despite the increasing chemical destruction of $NO_x$ in the cooler stratosphere (Stolarski et al., 2015). In summary, 1. higher efficiency of chemical ozone destruction due to the declining CFC levels and

2. stronger vertical transport will make EPP-$NO_x$ important for Antarctic upper stratospheric ozone over the coming decades. However, evolution of ozone depletion in the Antarctic lower stratosphere by EPP is more uncertain. More research, e.g., with idealized model experiment targeting impacts in future atmospheric state, is needed to quantify the EPP related ozone response at these altitudes.

*Data availability.* Data generated by SOCOL3-MPIOM simulations and used in this study is available at Zenodo repository

(https://doi.org/10.5281/zenodo.6553494), solar forcing for CMIP6 can be obtained from https://solarisheppa.geomar.de/cmip6.

*Author contributions.* V.M and A.S generated the research idea. P.A produced the SOCOL model outputs. V.M analysed the data and wrote the manuscript. All authors contributed to the analyses of the results and modification of the manuscript.

*Competing interests.* The authors declare no competing interest.

*Acknowledgements.* The research has been funded by the Norwegian Research Council under Contracts 223252/F50 (BCSS) and 300724

(EPIC). The simulations were performed on resources provided by UNINETT Sigma2 - the National Infrastructure for High Performance Computing and Data Storage in Norway.

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
