# Peer review of "Influence of energetic particle precipitation on Antarctic stratospheric chlorine and ozone over the 20th century"

_Atmospheric Chemistry and Physics, 2022_

## Author Comment (AC1)

**Reply to reviewers:**

We would like to thank both reviewers for careful reading of paper and valuable comments. For convenience, the original comments by reviewers are indicated below in **bold blue font**. Our response to each comment is given in normal black font.

Comments for anonymous referee (RC1):

**This paper investigates the influence of energetic particle precipitation on ozone and chlorine in the SH stratosphere over the 20th century by means of SOCOL3-MPIOM chemistry climate model simulations. EPP-induced NOx increases and associated ozone decreases were found to be in agreement with results of previous studies. A new finding is that EPP also induces substantial ClO decreases in the upper and mid stratosphere which reduces the ozone-depleting efficiency of EPP. In the lower stratosphere, EPP-induced ClO increases and ozone decreases were obtained at the end of the century while the opposite occurred before the period of high chlorine load. These results suggest a significant modulation of EPP-induced ozone loss by atmospheric chlorine which has implications for the future evolution of polar stratospheric ozone. This is a relevant topic and the paper is certainly suitable for publication in ACP.**

**The paper is well written, however, it fails short in convincingly identifying the chemical processes that are responsible for the EPP-induced ClO changes. Regarding the upper atmospheric response, the authors note in the abstract that the ClO decreases go along with increases in chlorine nitrate. A closer look at the absolute changes in the chlorine partitioning, however, suggests that most of the ClO is converted into HCl rather than into ClONO2.**

This is correct. We have now modified the abstract and parts of the text so that it is mostly HCl formation that counteracts the ClO decrease in the upper and mid-stratosphere. We have also included discussion about the HCl chemistry to the Introduction.

**Regarding the lower stratospheric response, the only explanation for the encountered ClO response is that "ClO is increased by activation of chlorine from the reservoirs" in the presence of PSCs. This is well known but does not explain why the ClO increase is enhanced by EPP. A possible reason for enhanced chlorine activation under EPP could be that the ClONO2-limited heterogeneous processing on PSCs in the Antarctic lower stratosphere is accelerated by the availability of more NOx and hence ClONO2.**

**It is also unclear why the lower stratospheric ClO response changes sign around the 80ths with increasing chlorine load. Is it possible that associated ozone depletion alters the Cly partitioning which could then modulate the ClO and O3 responses? Low ozone favors HCl formation and reduces ClONO2 by increasing the NO/NO2 ratio through the NO+O3 reaction (which then increases the rate of the ClO+NO reaction).**

**In summary, a more detailed analysis of the chorine partitioning in absolute terms (i.e. by use of line plots of the seasonal evolution of ClO, HCl, and ClONO2 from both EXP and REF simulations at 10 and 100 hPa levels for both low and high chlorine load conditions) would be very useful for identifying the responsible processes which, in turn, would significantly enhance the strength of this paper.**

Thank you for pointing this out. We have now included additional Figure 11, which shows 100 hPa seasonal variability of EXP and REF NO/NO2 ratio (a and b), climatology of HCl, ClONO2 and ClO in the

REF (c and d), and their EXP-REF difference (e and f) during 1924-1934 (pre-CFC era) and 1997-2007 (CFC era). These times were chosen since geomagnetic activity is roughly on the same level in solar cycles 16 and 23. Figure is also shown below. One can indeed see in subfigures a and b that the NO/NO2 ratio is larger during winter in the CFC era compared to the pre-CFC era, potentially (as you say) due to lower ozone levels and lesser NO+O3 reaction activity. Because there is more NOx due to EPP in general, reactions involving NOx and ClOx increase. In the pre-CFC era (subfigure e), low NO/NO2 ratio favors ClONO2 formation during midwinter. In the CFC era, NO/NO2 ratio is notably higher from June onwards, which limits ClONO2 formation and allows active chlorine (and ClO) to accumulate. This is also now discussed in the manuscript. See also reply to comment l190-193 below.

[Figure]

We also made similar figure for 10 hPa altitude, which is shown below. It shows very similar behavior in pre-CFC and CFC era for all variables (though magnitudes are obviously larger during the CFC era) and confirms that HCl balances ClO decrease in these altitudes. However, we feel that this figure is not necessary for the paper as these processes are explained sufficiently in the text. Thus, we have decided not to include it into the paper.

[Figure]

**l162-163: I agree that the ClONO2 decrease under EPP in mid-winter, seen in Fig. 8, suggests a Cly partitioning in favor of HCl by reaction R9. However, there is essentially no NO in the dark polar mid-winter stratosphere which could react with ClO. Although there could be a minor NO contribution from the the sunlit region, it is still striking that the ClONO2 decrease occurs in mid-winter and not in spring when sunlight (and hence NO) is available in the entire region.**

We have modified the discussion regarding HCl formation and give additional explanation via direct OH production by SPEs or mesospheric HNO3 as suggested by the other reviewer (see comments below for RC2). Following has been added: **"Alternative explanation to HCl formation due to EPP is via reactions (R10-R12) by SPE produced OH. Formation of mesospheric HNO3 due to EPP, and its subsequent descent to stratospheric altitudes during polar night with following photolysis might also play a role (Verronen and Lehmann, 2015)."**

**l168-169: Webster et al. (1993) looked at an Arctic winter which might not be representative for the Southern hemisphere. In any case, an explanation about *how* EPP reduces the HCl amount is missing.**

We added Molina et al. (1987) and following discussion: **"We suggest that excess NOx due to EPP leads to increased overall ClONO2 levels, which then enhances heterogenous reaction with HCl. While ClONO2 can be reproduced due to presence of excess NOx, HCl reduces significantly."**

**l180 "ClO-ClO catalytic cycle". Maybe "ClO dimer cycle" is more common.**

We changed this as **"ClO dimer cycle"**.

**l181ff / Fig. 9d: The EXP-RED TCO difference is negative throughout the winter/spring. This is in contradiction to the observational results of Gordon et al. (2021) who showed a TCO increase during SH spring (Oct-NOv) in high EPP years.**

It seems to be in contradiction. However, Gordon et al. (2021) time period is 2005-2017, while our simulation ends at 2008. One can also see that in Figure 9f EPP TCO effect is near zero or slightly above zero at the end of the simulation. We calculated Spearman correlation between geomagnetic activity and EXP TCO during Oct-Nov 1998-2008 and, while it was insignificant, it indeed was positive (R=0.21, p-value=0.54). Consequently, we added following: **"On the other hand, the EPP effect on TCO after 2000 returns to similar levels than before 1980 and even seems to be above zero at the end of the simulation. Gordon et al. (2021) showed positive correlation between geomagnetic activity and polar TCO in springtime (Oct-Nov) during 2005-2017. Spearman correlation between geomagnetic activity and EXP polar TCO during 1998-2008 is also slightly positive but insignificant (R=0.21, p-value=0.54)."**

**l190-193: The change of sign in the ClO response around the 80s is particularly interesting in Fig 10b. However, this is not discussed in the manuscript.**

Following the comment above and the new Figure 11, we discuss this as: **"To understand this change of sign in lower stratospheric ClO response to EPP, we analyze seasonal variability of 100 hPa NO/NO2 ratio, ClO, HCl and ClONO2 responses between low and high chlorine loading in Figure 11. Time periods of 1924-1934 (low chlorine) and 1997-2007 (high chlorine) were chosen since they represent roughly similar geomagnetic activity levels (see Figure 2a) during solar cycles 16 and 23, respectively. One can see that the NO/NO2 ratio in EXP starting from June is notably higher during the CFC era (Figure 11b) than in the pre-CFC era (Figure 11a). Potential explanation for this is lower ozone amount which reduces reaction (R7) and increases NO/NO2 ratio. There is generally more NOx available (due to EPP) to react with ClOx. In the pre-CFC era (Figure 11e), low NO/NO2 ratio favors ClONO2 reformation (reaction R8) during midwinter after heterogenous reaction between HCl and ClONO2 (Figure 11c). In the CFC era, higher NO/NO2 ratio limits ClONO2 reformation and allows active chlorine (and ClO) to accumulate (Figures 11d and 11f)."**

**l193-198: It is unclear how the discussion on GCR/EEP/SPE helps to understand the negative ozone response in the last two decades of the century. All these types of EPP produce NOx. The key questions are: Why is the lower stratospheric ClO response positive at the end of the century (it is negative in the middle and upper stratosphere...)? Why does it change sign with the onset of enhanced chlorine load?**

As seen above, change of sign in ClO response is now explained in separate paragraph. Discussion on GCR/EEP/SPE is meant to highlight that these lower stratosphere responses can also be related to forcing from above (EEP/SPE) rather than just direct effect from GCR.

**l198: "ClO seen in the lower stratosphere". Do you mean "ClO response seen in the lower stratosphere"?**

This is correct. It is changed as: **"ClO response seen in the lower stratosphere"**

**l213: "is more than expected". Do you mean "is more than unexpected"?**

Whole sentence is now rewritten as: **"Decadal geomagnetic activity decline in the late 20th century cannot solely account for this weaker ozone depletion by EPP."**

**l221: "We propose that this ClO increase can be explained by activation of chlorine from reservoir species ClONO2 and HCl." This is well known. What needs to be explained here is the positive ClO \*response to EPP\* after 1980.**

We now write: **"Activation of chlorine from reservoir species ClONO2 and HCl can be explained by heterogenous reactions on PSCs (Molina et al., 1987; Webster et al., 1993). We propose that during the pre-CFC era, excess NOx due to EPP enhances ClONO2 reformation via reaction (R8) after heterogenous reactions and leads to lower ClO levels due to EPP. However, during the CFC-era low ozone levels limit reaction (R7) between NO and ozone and leads to higher NO/NO2 ratio. This favors reaction (R9) between ClO and NO and limits reformation of ClONO2 between ClO and NO2 after heterogenous reactions, leading to excess active chlorine and ClO due to EPP."**

**l231: What do you mean with "ideal simulations"? Idealized model experiment? If yes, what kind of experiments?**

Yes, this is corrected as: **"idealized model experiment"**. Example of simulations could be using the same EPP forcing in the current CFC and greenhouse gas levels and in the future diminished CFC levels and/or increased greenhouse levels. This would allow us to study how the varying atmospheric state impacts EPP response, not only regarding ClO and ozone but other atmospheric variables too.

Comments for anonymous referee (RC2):

**This paper investigates the impact of energetic particle precipitation (EPP) forcing of chlorine species, and its consecutive impact on EPP-NOx driven stratospheric ozone loss. Ensemble model runs with and without EPP are carried out over the whole 20th century (1900-2008), a period with high solar activity and high chlorine loading in the second half of the 20th century. The impact of particle precipitation on NOy, HOx and ozone in the middle atmosphere has been studied in detail in a number of publications, but analyses of the impact of EPP on chlorine species are rare; the publication thus provides a new aspect. Of particular note is their observation that the high chlorine loading apparently had an impact on stratospheric ozone loss due to EPP, presumably by restricting both NOx- and ClOx-driven catalytic cycles due to the reaction of ClO with NO2. The inference is that in the coming decades, when the atmospheric chlorine loading will decrease, EPP ozone loss via NOx catalytic cycles will likely become more efficient. The paper is generally very well written, and the conclusions appear sound. However, conclusions could become more robust with a few more analyses, see suggestions below. Also it seems to me that the EPP ClOx mostly transfers into HCl, not ClONO2, and a more detailed discussion of this, and of possible pathways, would be useful.**

**Lines 54-56, R8 and R9: include and discuss pathways of HCl formation in the introduction, as this appears to be important as well: HOCl + Cl --> HCl + ClO; HO2 + ClO --> HCl + O3; OH + ClO --> HCl + O2, anything else? This works via an increase in HOx; EPP HOx is available during the particle precipitation in the (upper) mesosphere, but also possibly due to storage of EPP NOx and EPP HOx in the form of HNO3, which is transported down into the stratosphere during winter and there slowly photolyses, releasing both NOx and HOx (Verronen and Lehmann, GRL, 2015)**

Following the reviewer's comment, we have included discussion on HCl chemistry to the Introduction: "**Formation of hydrogen chloride (HCl) via reactions with HOx can also be important for EPP impact: ClO+OH->HCl+O2 (R10), ClO+OH->Cl+HO2 (R11), Cl+HO2->HCl+O2 (R12). EPP is known to produce HOx in the mesosphere (Verronen et al., 2011), and direct SPE production can reach upper stratosphere (Jackman et al., 2009). While HOx lifetime is too short to any EPP indirect stratospheric HOx, formation of HNO3 in reaction between NO2 and OH, and its subsequent descent to stratospheric altitudes during polar night might also play a role (Verronen and Lehmann, 2015). Finally, reaction Cl+CH4->HCl+CH3 (R13) can also be important following the reaction (R9) (Brasseur and Solomon, 2005).**"

**Line 89: wouldn't it be more exact to use only data from REF for the estimation of the significance? Then (EXP-REF) would be tested against the variability of REF, which seems to be more to the point.**

We define our null hypothesis so that there is no difference between EXP and REF. This is the same null hypothesis as with a two-sample t-test. To do this we need to compare against a so called pooled standard deviation (both EXP and REF), similarly as in a two-sample t-test. The motivation to use MC simulation instead of a t-test is that we don't need to worry about temporal autocorrelation or other underlying properties of the data when calculating the significance, but they are automatically considered in MC simulation. This is not the case in a standard t-test.

**Line 104: maybe you could say a few more words about the content and meaning of Fig 2. Figure 3 – for better readability, please include ticks for 10 hPa and 0.1 hPa for the vertical axis. Same for Fig 4 and following.**

This is a good point. We have included some discussion concerning Figure 2 to the Data and Methods section. Figures 3-5 now have also ticks for 0.1 and 10 hPa.

**Line 117: if the mesospheric ozone depletion is due solely to in-situ EEP HOx production, why is it stronger in the Southern hemisphere? Doesn't the difference between Northern and Southern hemisphere imply a dynamical/long-lived component in the mesospheric ozone depletion as well? Possibly HNO3 formation/photolysis?**

This is an interesting observation, there seem to be somewhat more mesospheric ozone depletion during mid-winter in the southern hemisphere than in the northern hemisphere. We have included following discussion: **"However, mesospheric ozone depletion seems to be slightly stronger in the southern hemisphere than in the northern hemisphere, which implies an additional dynamical or long-lived component in the southern hemisphere. This can be explained by southern hemispheric polar vortex forming earlier and being more stable (Andersson et al., 2018) and/or via HNO3 formation in the mesosphere (Verronen and Lehmann, 2015). Thermospheric NO in the model is prescribed with a semi-empirical model, which on average has more NO entering the mesosphere in the southern hemisphere (Funke et al., 2016)."**

**Line 120-121 and following discussion of lower stratosphere ozone anomaly: the positive ozone anomaly covers nearly the whole lower stratosphere, from high Southern to high Northern latitudes, with the exception of the polar winters, when the anomaly turns sign. I mid-and low-latitudes, this positive anomaly is interpreted by the authors as a GCR impact (line 123), and this appears likely. However, I would argue based on the spatial/temporal evolution of this signal that this GCR signal extends from pole to pole, but is overwritten by the auroral signal indirect effect during polar winter.**

We agree. We now say: **"There is also a weak but significant ozone response around 100 hPa altitude covering all latitudes. It is positive and significant in the low latitudes all year and in the high latitudes during summer, but significantly negative during winter/spring, at least in the Antarctic. This Antarctic lower stratosphere response is also in agreement with Rozanov et al. (2012) and Damiani et al. (2016). Positive weak ozone response in the lower stratosphere all year is a consequence of GCR (Calisto et al., 2011; Jackman et al., 2016), while it is likely dominated by indirect EPP effect in the high latitudes during winter."**

**Line 129: Are these corresponding to the negative NOx anomalies? And, is there a corresponding anomaly of ClONO2?**

We have modified the interpretation of lower stratospheric response and added a new Figure 11 to the paper. See also comments above for reviewer one (RC1).

**Line 135: Over hundred percent --> more than a hundred percent**

Corrected.

**Line 143: … as can be seen in Fig 2a --> by comparing with the Ap index shown in Fig 2a. However, it would be better to provide some hard numbers here to substantiate this statement, e.g., by providing a correlation coefficient (preferably from some ordered, nonlinear method – rank? – not Pearson) between NOx and Ap.**

We have now included Spearman correlation value between 10-20hPa NOx and Ap index (R=0.73, p-value practically zero for EXP in Fig. 7b and R=0.84 for EXP-REF in Fig. 7c).

**Fig 7 b and e, as well as following figures – can you provide error bars due to ensemble variability for the timeseries?**

We tried this with 95% confidence intervals (±Standard error of the mean*1.96), but because the intervals are fairly small, they are barely visible. We decided to leave original figures since adding confidence intervals does not provide much additional information for the reader. See also example figures below.

[Figure]

**Fig 7 c and f: the lines appear to be anti-correlated – are they? E.g., provide correlation Coefficient**

Correlation between EXP-REF NOx and EXP-REF ClO is -0.48, pvalue<0.01. In the text we provide Spearman rank correlation value between Ap index and EXP-REF ClO (R=-0.46, p-value<0.01). See also comment below for Line 151.

**Line 147: Loss of … as this is a decrease relative to the reference scenario, I wouldn't call**

**it "loss", which would imply chemical loss**

Good point. We now say: **"More than 15 percent ClO decrease in EXP relative to REF continues well into spring extending down to 40 hPa."**

**Line 151: seems to be anticorrelated --> just provide the correlation coefficient**

We now provide Spearman rank correlation value between Ap index and EXP-REF ClO (R=-0.46, p-value<0.01).

**Line 154: response --> response to EPP forcing**

Now corrected.

**Lines 154-156: and possibly because NOx is bound in PSCs in the form of HNO3?**

As stated above, we have modified the interpretation of lower stratospheric response and added a new Figure 11 to the paper. See also comments above for reviewer one (RC1). These processes are discussed later in the paper and following statement is placed here: **"Influence of EPP on these processes in the lower stratosphere is discussed in more detail below."**

**Line 158-159: Substantial increase compared to what – EXT to REF, or to the beginning of the model periods?**

We wrote this now in more detail: **"Substantial increase of ClONO2 in EXP relative to REF is seen in the upper stratosphere during winter (Figure 8a)."**

**Line 161: the ClONO2 amount is not negative in your model runs (one hopes), it is less than in the REF scenario without EPP.**

This is true, we now say: **"Between 3 and 10 hPa, ClONO2 amount is less in EXP than in REF during mid-winter."**

**Line 163: also strengthened by the fact that the ClONO2 difference in absolute numbers seems to be much smaller than the HCl difference. I think you could explore this in more detail. Do you really think this is due mainly to CH4 + Cl?**

We have added following discussion: **"Alternative explanation to HCl formation due to EPP is via reactions (R10-R12) by SPE produced OH. Formation of mesospheric HNO3 due to EPP, and its subsequent descent to stratospheric altitudes during polar night with following photolysis might also play a role (Verronen and Lehmann, 2015)."** See also figure above for second comment to RC1.

**Line 173: again, just provide a correlation coefficient**

We now give correlation values for period 1900-1960 (R=-0.65, p-value<0.01) and for period 1961-2008 (R=-0.07, p-value=0.65).

**Figure 8 c and f: are the lines anticorrelated? Are they correlated/anticorrelated to the Ap index shown in Fig 2a?**

Spearman correlation between lines 8c and 8f is 0.25 (p-value=0.02). Correlation between Ap and 8c is practically zero over the whole time period (R=-0.01, p-value=0.95), and between Ap and 8f is 0.48 (p-value practically zero). We also added following: **"Positive correlation between annual geomagnetic activity and Figure 8f data (R=0.48, p-valu<0.01) also implies that ClO is mostly accounted by excess HCl, instead of ClONO2 (correlation between geomagnetic activity and Figure 8c data is -0.01 over the whole time period)."**

**Line 215: We find a significant decrease of stratospheric ClO … relative to a model run without EPP impact / due to the EPP impact**

This is corrected as: **"We find a significant decrease of stratospheric ClO due to the EPP impact in the same altitude…"**

**Line 219: ClO abundances decrease … relative to a model run without EPP .. by …**

We corrected this as: **"ClO abundance increases relative to the model run without EPP by more than 5 percent during winter after 1980."**

**Line 220: Why is this negative before 1980?**

This is now explained as following: **"The seasonal emergence of this ClO response is consistent with the formation of PSCs in the Antarctic and occurs slightly before the depletion of ozone at the same altitude. Activation of chlorine from reservoir species ClONO2 and HCl can be explained by heterogenous reactions on PSCs (Molina et al., 1987; Webster et al., 1993). We propose that during the pre-CFC era, excess NOx due to EPP enhances ClONO2 reformation via reaction (R8) after heterogenous reactions and leads to lower ClO levels due to EPP. However, during the CFC-era low ozone levels limit reaction (R7) between NO and ozone and leads to higher NO/NO2 ratio. This favors reaction (R9) between ClO and NO and limits reformation of ClONO2 between ClO and NO2 after heterogenous reactions, leading to excess active chlorine and ClO due to EPP."**

**Line 230: In principle, I agree with this conclusion, but find "crucially" maybe a bit too strong / confident.**

We removed the word **"crucial".**

---

## Author Response (AR2)

**Reply to reviewers:**

We would like to thank both reviewers for careful reading of paper and valuable comments. For convenience, the original comments by reviewers are indicated below in **bold blue font**. Our response to each comment is given in normal black font.

Comments for anonymous referee (RC1):

**The authors have done a very good job in addressing my comments. Overall, I think the paper is now ready for publication. There are, however a few minor issues and typos which should be addressed in the final version:**

**l 56 ClO + NO2 → ClONO2 should read ClO + NO2 + M → ClONO2 + M**

Reaction now includes a non-reactive molecule M.

**l 69 ...reach THE upper stratosphere....**

This is now corrected.

**l 69 ...too short FOR any....**

Corrected using "for".

**l138 maybe clarify that the stronger SH vortex is less diffusive and hence O3 depleted vortex air remains better confined.**

We added following: **"This can be explained by southern hemispheric polar vortex forming earlier and being more stable and less diffusive, which confines ozone depleted air more efficiently."**

**l151 by THE indirect EPP effect**

This is now corrected.

**l190-201 This paragraph is now a bit confusing as it mixes up ClONO2 and HCl responses at different altitude ranges. I would here restrict to the discussion of responses of both species in the 3-10 hPa range, concluding that ClO is buffered mostly into HCl rather than ClONO2 in EXP, and leave the discussion of 10-20 hPa responses for the next paragraph. Also, from Fig 2 of the response letter, it becomes clear that the driving process for ClO->HCl conversion is the reaction of ClO+NO (ClO decrease in panels e and f correlates very well with the NO/NO2 ratio in panels a and b, indicating that ClO+NO rules both the ClO and the NO/NO2 response). This should be made clear.**

We separated the discussion into two paragraphs. We also highlighted slightly more the explanation via reaction R9.

**l204-205 I think that ClO is released by the heterogeneous reaction of ClONO2 and HCl. Since there is way more HCl compared to ClONO2 in the SH lower stratospheric polar vortex in the CFC era, this reaction stops when all ClONO2 is processed. However, under EPP there is a steady supply of ClONO2 via enhanced NOx which keeps this process running, resulting in less HCl, more ClONO2, und finally more ClO via processing on PSCs.**

We modified the text as: **"ClO is mostly released by consuming HCl which recovers slowly to initial values, while ClONO2 levels can rebuild fairly quickly and to excess levels of its initial values (Molina et al., 1987; Webster et al., 1993), especially in the presence of additional NOx due to EPP. This means that EPP keeps the heterogeneous processing between HCl and ClONO2 running and results in less HCl, more ClONO2 and finally more ClO via processing on PSCs."**

**l219 ...via THE ClO dimer cycle**

Corrected with adding "the".

**l240 increases THE NO/NO2 ratio**

This is now corrected.

**l241-242 maybe you can add that heteogeneous processing on PSCs is HCl-limited in the pre-CFC era while being ClONO2-limited afterwards (due to the O3 impact on the NOx partitioning), compare Fig 11 panels c and d.**

**l242-243 I think that the principal mechanism during the CFC era (ClONO2-limited heterogeneous processing) is that the NOx supply under EPP conditions keeps the ClONO2+HCl reaction running, resulting in ClO increase (see above).**

We included following: **"One can see that the NO/NO2 ratio starting from June is notably higher during the CFC era (Figure 11b) than in the pre-CFC era (Figure 11a). Potential explanation for this is lower ozone amount which reduces reaction (R7) and increases the NO/NO2 ratio. There is generally more NOx available (due to EPP) to react with ClOx. In the pre-CFC era, low NO/NO2 ratio favors ClONO2 reformation (reaction R8) after heterogeneous reaction between HCl and ClONO2 (Figure 11c) resulting with less ClO and more ClONO2 due to EPP (Figure 11e). Thus, heterogeneous processing on PSCs is HCl-limited in the pre-CFC era (Figure 11c). In the CFC era, higher NO/NO2 ratio limits ClONO2 reformation and allows active chlorine (and ClO) to accumulate (Figure 11d). Heterogeneous processing is thus ClONO2-limited. While EPP related ClONO2 production in the CFC era is also slowed due to higher NO/NO2 ratio, it is still net positive over the whole winter (Figure 11f). This excess ClONO2 due to EPP is able to keep the heterogeneous processing running, resulting in ClO increase and considerable HCl decrease over the whole winter (Figure 11f).**

**l281-282 see point above.**

We modified the discussion as: **"We propose that during the pre-CFC era, heterogeneous processing on PSCs is HCl-limited and excess NOx due to EPP enhances ClONO2 reformation via reaction (R8) leading to lower ClO levels. However, during the CFC-era low ozone levels limit reaction (R7) between NO and ozone and leads to higher NO/NO2 ratio. This limits the ClONO2 reformation and makes heterogeneous processing ClONO2-limited. While EPP related production of ClONO2 is also slowed in the CFC era, it is still net positive over the whole winter. EPP is thus able to keep heterogeneous processing running and results in increase of ClO and substantial decrease of HCl. These results imply that EPP has significantly modulated chemical processes responsible for ozone hole formation."**

Comments for anonymous referee (RC2):

**Figures 7 b and e and following (Figs 8 b and e; 9 b and e; 10 a and c), response to reviewer 2: I still think showing the error bars makes sense, even if they are very small - they are visible mostly in the example shown, and clearly show the differences are significant.**

Following the reviewer's comment, we decided to include the error bars to Figures 7-10 (Subfigure b and e).

**The link to the SOCOL-MPIOM3 data provided in the data availability section did not work for me (June 5, 2022) - please check.**

We checked this and the doi link works for us (https://doi.org/10.5281/zenodo.6553494). Maybe it is some issue with the pdf reader. Alternative direct link to zenodo repository is https://zenodo.org/record/6553494.